# Much More Than Food: The Malaysian Breakfast, a Socio-Cultural Perspective

Jean-Pierre Poulain [1,2,3,4,*], Elise Mognard [1,2,3,4], Jacqui Kong [1,3,4], Jan Li Yuen [1,2,3,4], Laurence Tibère [1,2,3], Cyrille Laporte [1,2,3], Fong-Ming Yang [1,3,4], Anindita Dasgupta [3,4], Pradeep Kumar Nair [3], Neethiahnanthan Ari Ragavan [3,4] and Ismail Mohd Noor [3,4,5]

1  Chair "Food Studies: Food, Cultures & Health", Taylor's University Malaysia, Subang Jaya 47500, Malaysia
2  Centre d'Études et de Recherche: Travail, Organisation, Pouvoir (CERTOP) UMR CNRS 5044, Université de Toulouse, 31000 Toulouse, France
3  Center for Asian Modernisation Studies (CAMS), Taylor's University, 47500 Subang Jaya, Malaysia
4  Faculty of Social Sciences and Leisure Management, Taylor's University, Subang Jaya 47500, Malaysia
5  Centre for Community Health Studies, Faculty of Health Sciences, Universiti Kebangsaan Malaysia, Kuala Lumpur 50300, Malaysia
*  Correspondence: jean-pierre.poulain@univ-tlse2.fr

**Abstract:** Using secondary analysis of data from the Malaysian Food Barometer (MFB), this article highlights ethnocultural dimensions and social functions of breakfasts in the Malaysian population. MFB uses a 24-h dietary recall that lets the interviewee give the name of the food intake. It shows that breakfasts from the Asian food register dominate with 50.7% (Malays, 50.4%; Indians, 51.9%; Chinese, 47.6%; non-Malay Bumiputra 50.1%), whereas 26.1% eat a westernised breakfast and 17.6% eat no breakfast. If we add those who just have a beverage, 20% do not eat a "proper" breakfast. The Asian breakfasts are characterised by including cooked dishes. These sometimes require real craftmanship to prepare. Therefore, they are mostly purchased outside and consumed either at home, at the workplace, or outside, in restaurants or food courts, such as "mamaks" or "nasi kandar ". Breakfast dishes can be attached to the food culture of the three main ethnic groups of Malaysia, but the boundaries between breakfast cultural styles are fluid and there is a sort of pooling of the breakfast dishes. This porosity of the boundaries between culinary styles is one of the main characteristics of Malaysian breakfast culture. It is so important that when asked, "What could represent Malaysia the best for submission to UNESCO's intangible heritage list?", the sample of a national representative population places two breakfast dishes first (nasi lemak and roti canai). This knowledge of the ethno-cultural dimensions of breakfast will help public health nutritionists and policymakers consider cultural characteristics and avoid the risk of a (non-conscious) neo-colonial attitude in promoting western style breakfasts. However, bearing in mind the influence of the British colonisation, the so-called westernised breakfast could also be considered as part of a cosmopolitanised breakfast culture. Finally, the understanding of breakfast culture will feed the debate around, and the progress towards, sociocultural sustainable healthy diets.

**Keywords:** anthropology of food; breakfast pattern; food day; food culture; sociocultural sustainable healthy diet; sociology of food; breakfast style

## 1. Introduction

Breakfast, as the first meal of the day, varies considerably over time and space. Within the same culture, it changes over time, in terms of content, timing, form of socialisation and position of the eater in the social hierarchy. Between cultures, it varies in the same ways but with even greater amplitudes. The experience of a breakfast buffet in an international hotel offers a concrete vision of this diversity. There are substantial differences between the full English breakfast with eggs, bacon, fried potatoes, toast, tea and more, and the French breakfast with coffee, bread, butter and jam; between the Indian breakfast with roti canai,

dals, masala puris, upuma, idlis and dosas, and the Chinese breakfast based on congee (zhōu, 粥), steamed stuffed buns (bāozi, 包子), wontons (húntún, 馄饨/yúntūn, 云吞) and Guilin rice noodles (guìlín mǐfěn, 桂林米粉).

Contemporary Malaysia is a multicultural society with several ethnic groups: Malays, Chinese, Indians and non-Malay Bumiputra. The latter also referred to as "sons of the soil" refers to the indigenous inhabitants of Malaysia. It officially includes the Orang Asli - aboriginal populations of the peninsula - and the natives of Sabah and Sarawak) [1–4]. All these groups have their own food cultures, based on ancient traditions, taboos and prohibitions. These traditions have drifted further over time with the diasporic phenomenon. Moreover, Malaysian food cultures and the conception of breakfast have been influenced by the British colonisation. In this country, food cultures are quite diversified and the "traditional" conceptions of breakfast vary considerably. It is in this socio-cultural context that modernisation of food occurs.

Food modernisation can be characterised by simultaneous and contradictory trends that affect the relations to food [5–7]. The first one is the homogenisation and westernisation of practices under the influence of globalisation of food markets and the emergence of transnational actors in agro-food industries and restaurant services. Pertaining to breakfast, the effects are the diffusion of products coming from an occidental food culture, (such as breakfast cereals and biscuits or breakfasts offered in fast food restaurants) and the dissemination of the western vision of breakfast through nutritional recommendations [8].

The second trend is the dissemination of nutritional culture, which has been described as "nutritionalisation" [9]. In a context of an obesity epidemic and of the rise of non-communicable diseases (NCDs), not only policymakers, but also actors of the food system, promote the idea of "healthy food environments" [10,11] and the implementation of a "nutritional" conception of food habits. However, food modernisation is also and at the same time characterised by the idea that food and food practices are a space of expression of cultural identities. Therefore, culinary and food practices are viewed as a heritage that must be protected and revitalised. Facing environmental concerns because of climate change, the idea of more sustainable food systems and diets is increasingly accepted [12–14]. The purpose of this article is to highlight the ethnocultural dimensions and the social functions of breakfasts in the Malaysian population. This knowledge will help public health nutritionists and policymakers deal with cultural characteristics and avoid the risk of a (non-conscious) neo-colonial attitude. Finally, the multicultural Malaysian society will be a suitable empirical field to study in detail the distinction between social and cultural dimensions of food and to feed the debate on the conception of sociocultural sustainable healthy diets [13,15,16].

## 2. Theoretical Background and Methodology

This article is based on a mixed method including secondary analysis of quantitative data from the Malaysian Food Barometer (MFB), expert interviews and participant observations. The MFB database is open access and can be consulted online [17]. The theoretical framework, questionnaire, data collection and sampling methods have been presented in several publications [6,17–19]. However, for this paper, strengthening certain theoretical and methodological points related to breakfasts proved necessary.

The traditional nutrition surveys assume that eating behaviour is the result of the eater's decision process. In this perspective, differences in the behaviours come from individual preferences and the arbitration between advantages and disadvantages. Rational choice theory postulates that when individuals are well informed, or, more specifically, aware of these advantages and disadvantages, they behave in a way to optimise their advantages. Thus, eating well would be a matter of relevant information. Dispelling misconceptions and replacing them with correct science-based information is the rhetoric of nutrition discourse.

The theoretical perspective of MFB breaks with the dominant reading grid in classical nutrition studies based on the theories of rational choice or programmed action. The ap-

proach postulates that a significant part of eating behaviour is not the object of reasoning by eaters and that a substantial part is "unthought". Eating practices result from socially and culturally predefined behavioural patterns and routines supported by social norms [20–23]. Social influences do not only stem from social position, education, gender, income, tastes and preferences, but also from the infrastructure of practices that are the meals and food patterns, incorporating social and cultural influences [24]. The MFB database is primarily designed to study these determinants and their modalities of action. By focusing on these patterns, norms and practices, MFB explores some of nutritional surveys' blind spots.

Let us take the example of an individual who "decides" to go for breakfast. Depending on the food culture they belong to, their reasoning and the choices they must make will differ widely. In some contexts, they will not even decide in the sense of a 'reflective act'. Take, for instance, the case of a French and a British person. The room of decision will first and foremost be within structures like "hot drink" + "bread or pastry" + "butter, jam/spread". The choice will therefore be made within the structure of the hot drink; the bread, toast, biscuits, pastries; the butter and jam/spread. Other breakfast models are possible in this culture but not dominant. Variants can include more foods such as cheese, cold cuts, cereals, fruit juices. They are the result of several phenomena. First, nutritional discourse in favour of a "hearty" breakfast, to which the food industry often follows suit, when it does not precede it. Second, the aggregation of the French breakfast with the "case croute". This "small meal" is taken by people who start work early in the morning to complement a rather light breakfast. This combination of two morning intakes undoubtedly explains the large proportion of very simplified breakfasts ($\pm$ 20%) observed in the French population [25]. The British person's choices will be made within the framework of a more complex structure which includes eggs, beans, sausages, mushrooms, tomatoes, toast, etc. These "breakfast structures" are visible on the breakfast menus of international hotels, under the names "continental breakfast", "Full English breakfast" and the like.

In Malaysia, where different food cultures coexist, Malay, Indian and Chinese breakfasts occur with different structures and different dishes. As a result of colonisation, the British structure is also active. In addition, some flagship products slip into food spaces and some interbreeding takes place. The "nasi lemak", for example, which is a traditional Malaysian breakfast, is consumed by all Malaysian ethnic groups, with different frequencies. What happens with breakfast also applies to all other meals.

Food patterns are not only "used" by eaters but also by the household members, who shop, cook and prepare meals, and by all those who directly or indirectly participate in the production, processing and distribution of food products and services. They constitute the cognitive infrastructure that allows these actors to coordinate and contribute to the functioning of the food system. There is indeed room for decisions, but these are embedded in the multiple patterns and categories of food cultures. Food patterns are the infrastructure on which decisions are made. Eating behaviours operate in a self-evident way, almost in an unthoughtful, unreflective way [21,23]. They are also the result of social interactions, of which food is only a support, (such as invitations to refill, to taste, to try, gift counter-gift, encouragement, etc.). In addition, meals have social functions and play roles in the social life by being a time where social links are maintained and social identities expressed and affirmed.

This theoretical framework thus distances itself from the classic nutritional approach explicitly included in the theories of rational choice. For the "Sociology of Eaters" [7,26,27], the decisions of eaters are embedded in culturally defined scenarios of action. Thus, there are many decisions, but they take place in a restricted and socially determined space of freedom. Therefore, a priority for research is to uncover these meal patterns. For that we need to go back to the definition(s) of a meal as a combination of several foods and drinks taken at a certain period of the day and in a more or less socialised context [28,29].

The second point that we need to strengthen the theoretical frame is the place of breakfast in the food day. The classical nutrition surveys are based on the implicit assumption that human beings eat three meals a day and that one of these meals is breakfast. Starting

from this implicit postulate, almost all the nutritional data processing software require that the question of the 'name of the meals' is answered first in the organisation of data entry. This method leads to subsequently identifying the individuals who do not eat one of the meals as a "skipper". These people will then be seen as needing nutritional advice, centred on the "Western" vision of three meals a day.

Data collection tools, whether 24-h dietary recall or multi-day (3 or 7) diaries, overwhelmingly use meal names as input to the questions. In the self-administered questionnaires, the names of the meals are recorded in the notebook and constitute structuring elements of the response and the recall. When the researchers using these tools claim "self-designation" by the respondent, they mean that it is the respondent who registers content (food and beverages) in the "breakfast" category. It is therefore not the same method as that used in the "Food Barometers", in which it is food intakes (food contacts) that are the input into the reconstruction of food days. It is to avoid, among other things, this type of culture-centrism that the Food Barometers philosophy has been designed. In the MFB, the dietary recall of 24 h follows the flow of the day, from the time of waking up until the time of going to bed. It identifies "food intakes" (all meals and other intakes) and only after this identification is the name of the food intake given, by the interviewee himself.

So, how many meals are consumed in a day and what is the first meal of the day? On those two questions the literature on the socio-anthropology of food shows that there are different food models all over the world based on variable numbers of meals (a meal could be defined and described as a food intake with five main dimensions: 1. The time dimension takes into account both the time of day - time setting - and the duration of food intake. 2. The structure of the intake (solid, liquid, combined). 3. The spatial dimension - distinction between out-of-home and at-home. 4. Social synchronisation refers to the meeting points in the schedules of the different social actors, allowing meals to be shared. 5. The social environment - alone or in company [30,31]) and intakes—2, 3, 4 or more [28,32–34]. The nutrition and public health literature relating to the ideal number of daily intakes does not seem able to arbitrate the question.

## 3. Malaysian Breakfasts, Elements of Linguistics

The following definitions were derived from dictionaries and online interviews with experts and native speakers. This method was deemed the most suitable approach to obtain localised information and insight. The criteria for selecting interviewees were native speakers or at least speakers of the language for several years, speaking from a young age, having a family background with that language and/or being an expert in the field of linguistics, culture or food.

In MFB the name of the food intake is given by the interviewees themselves. First of all, the word "breakfast" is used by an overwhelming majority even when they speak *Bahasa Malaysia* (>90%). To explore the understanding of the linguistic roots of the local words or expressions, we prepared a breakfast exploration dictionary and interviewed seven experts (see Appendix A). In Malay, the definition of a breakfast refers to three main significations.

The first one is linked with the temporal organisation of the food day. Breakfast in this perspective is the first meal of the day, "makan pagi" (morning meal). The second signification is breaking the fast after an overnight sleep. Here, the emphasis is on the fact that it is the first meal or food intake after a period of fasting. Both significations include the idea that breakfast is a meal. What defines a meal? A certain quantity of food, combination of food and drinks, ritualisation, time implementation. All these characteristics are socially defined. The last definition introduces the idea of filling the belly, alas, lapik, lapisan or "sesuatu yang dibuat alas atau lapik" (something made as a layer), "makanan pagi sebagai alas perut" (morning food as layer for the belly). The linguistic exploration of the way breakfast is expressed is shown in Table 1.

**Table 1.** Linguistic Exploration of the way to say "breakfast".

| | First Meal of the Day | Breaking the Fast | Filling the Belly |
|---|---|---|---|
| Malay | Makan pagi (to eat in the morning) Sarapan Sarapan pagi | Juadah Pagi (Food served in the morning) | Alas, lapik, lapisan or "sesuatu yang dibuat alas atau lapik" (Something made as a layer), "makanan pagi sebagai alas perut (Morning food as layer for the belly) |
| Chinese Mandarin | 早餐 Zǎocān (Meal eaten in the morning, the first of the day) Zǎo—morning. Cān—meal 早点 Zǎodiǎn (Morning meal) | 早饭 Zǎofàn (Morning rice) Zǎo—morning. Fàn—rice | |
| Cantonese | 早餐 Zou can (Morning meal) | 早饭 Zou fan (Morning rice) | |
| Hokkien | Char1chan1 (Early meal) char1-early chan- meal Char1tnui3 (Early meal) char1-early tnui3-meal Chiak2 cha1ki1—(Eat morning) Chiak- to eat Cha1ki1-morning | | |
| Indian Tamil | | காலை உணவு Kālai uṇavu (Morning food) | Pesiare (alas perut) |

Sources: Expert interviews and Pleco dictionary, [35,36].

In Mandarin, Cantonese and Hokkien, two expressions put the emphasis on the morning associated with the idea of a meal: "meal of the morning", or the idea of eating: "eating in the morning". The second expression puts the emphasis on the food itself: "morning rice'. In Tamil we found also 'morning food" and the idea of 'filling the belly'.

## 4. Malaysian Breakfast Styles

From a composition point of view, the first meals of the day in Malaysia can be broken down into four main categories. More than 50% of the population eat an Asian type of breakfast, 26.1% a Westernised style and the remaining are divided as follows: 4.3% of people that just take a small snack (a small cake or a fruit) and 17.6% who don't eat in the morning. (Table 2, comprehensive table of data analysis is provided in Appendix B). Among the Malays, 16.3% individuals do not eat breakfast; 20.7% if we add the ones who just have a beverage. This is the highest level of the Malaysian population. This percentage is 8.6% for Chinese (15.8% if we add the ones who only take a beverage) and 6.0% for Indians (12% when those who only have a beverage are added).

**Table 2.** Styles of breakfast (%) and ethnicity (N = 2000).

| Ethnicity | Asian Style | Western Style | Other Simplified Forms | No Breakfast | Beverage Only | Total |
|---|---|---|---|---|---|---|
| Malay (N = 1176; 58.8%) | 50.4 | 24.4 | 4.4 | 16.3 | 4.4 | 100 |
| Indian (N = 133; 6.7%) | 51.9 | 30.8 | 5.3 | 6.0 | 6.0 | 100 |
| Chinese (N = 498; 24.9%) | 50.1 | 30.3 | 3.8 | 8.6 | 7.2 | 100 |
| Non-Malay Bumiputra (N = 193; 9.7%) | 57.0 | 29.0 | 3.6 | 4.7 | 4.7 | 100 |
| Overall | 50.7 | 26.1 | 4.3 | 12.7 | 5.2 | 100 |

The meals consumed at breakfast can be attributed to an ethnic food style. To Malays, the different variants of "nasi lemak", "nasi goreng", "kuih"; to Indians, the different variants of "roti canai", "chappati"; and to Chinese, "noodle soup", "fried rice", "dim sum". To further explain, 'Nasi lemak' is a dish consisting of fragrant rice steeped in coconut milk and pandanus leaf. The result is an iconic Malaysian dish of fragrant, pillowy rice served along with condiments such as deep-fried anchovies and peanuts, hard-boiled egg, slices of fresh cucumber, and a sweet or spicy 'sambal'—a thick sauce made from a blend of aromatics, spices and oil. 'Nasi goreng' is a literal translation of 'fried rice'. In Malaysia, there are many varieties of nasi goreng due to the multicultural and multi-ethnic society of Malaysia. The most common types of nasi goreng feature common ingredients of rice, vegetables, protein and egg. Chillies and type of sauce/s used depend on the particular 'style' and recipe of nasi goreng. 'Kuih' means 'cake' in the Malay language, and refers to any type of small, bite-size sweet or savoury snack 'cake'. There is a wide variety of 'kuih' in Malaysia, and they are usually made from coconut milk, glutinous rice or rice flour. They are usually bouncy or chewy in texture and can be eaten for breakfast, snack/tea time or dessert, depending on the type of 'kuih'. Again, the variety of 'kuih' is dependent on the specific ethnic origin of the recipe. 'Roti canai' is a type of flatbread dish in Malaysia, created from a dough made with flour, water and fat (usually ghee). It is also known as 'roti prata' and is usually eaten as a snack item, or for breakfast or late-night suppers. It is usually accompanied by 'dal', a lentil gravy, or you can choose to eat it with various curry sauces or filled with fillings such as fried egg or onions. 'Chappati' is a type of unleavened flatbread of Indian origin. It is less rich than 'roti canai' as it is made from wheat flour, salt and water, and thus seen as the healthier alternative. The accompanying sauces or gravies and fillings are similar to those of 'roti canai'. 'Dim sum' is a Chinese meal consisting of plates of small items of food, such as dumplings, snacks or buns. The dumplings range in a variety of fillings such as seafood, meat and/or vegetables. Other popular 'dim sum' dishes also include rice paper rolls, stewed/braised chicken feet, pork buns, glutinous rice, taro or turnip cake and sweet pastries such as egg tarts, custard buns and red bean buns for dessert. 'Dim sum' is usually eaten during brunch and lunch hours and served with tea.

In an undifferentiated Asian category, are grouped the "fried mee" and their derivatives. The distribution of breakfast styles in the Malaysian population shows that breakfasts from the Asian food register dominate, with more than 50% (for Malays, 48.7%; Indians, 51.9%; Chinese, 47.6%). These breakfasts have the characteristic of including cooked products. They sometimes require real technicality and are, for a significant part, purchased outside and consumed either at home, at the workplace, or even outside in catering places such as "mamaks". These establishments are open 24 h a day but serve breakfast dishes in the time window from 6 to 11 am. Many restaurants or stalls in the street offer "nasi lemak" wrapped in banana leaves and paper that can be taken away easily. The Malaysian breakfast is largely culinary and consists of cooked dishes that require some

preparation time. In addition, these dishes, although specific to breakfast, are also served in restaurants in modernised versions and at different mealtimes. One of the consequences of these styles of "culinary" breakfasts is that a high proportion of these meals are eaten away from home (43%), either in restaurants or at work (after having purchased the dishes in question on the way there). Thus, an important part of the culinary decisions escapes the control of the eaters.

More than one Malaysian in four among the ones who eat breakfast opt for a westernised style breakfast or one including 'Western' products. 'No breakfast', as well as westernised breakfast styles, are linked with modernisation (the modernization index constructed in the Malaysian Food Barometer aims to reflect the transformations faced by the Malaysian society in terms of urbanisation, changes in the social stratification with the emergence of a middle class (proxy: income levels), social mobility (proxies: education attainment and income dynamics) and socio-demographic transition (proxy: size of household)) and urbanisation.

### 4.1. Malaysian Breakfast as a Multicultural Phenomenon

The boundaries between breakfast styles are open and show commonality. This porosity of the boundaries between culinary styles is undoubtedly one of the main characteristics of Malaysian food culture. It is reflected in the fact that, while belonging to a subset of the Malaysian population, all these various products are clearly part of the Malaysian breakfast range (Table 3, comprehensive table of data analysis is provided in Appendix B).

**Table 3.** Typical dishes and ethnicity in Malay, Indian and Chinese Malaysians (N = 1443).

| Ethnicity | Typical Dishes | Other Typical Dishes | Undifferenced Asian Dishes | Total Asian Dishes | Western Dishes or Products |
|---|---|---|---|---|---|
| | 18.3 | 20.5 | 11.5 | 50.4 | 24.4 |
| Malay | Nasi Lemak 11.8<br>Kuih 4.9<br>Nasi Goreng 1.5 | Roti Canai 6.5<br>Chapati 4.1<br>Noodle soup 6.6<br>Fried rice 3.3 | | | Bread 11.8<br>Biscuit, croissant 6.3<br>Sandwich 4.4<br>Cereal 1.9 |
| | 10.5 | 30.8 | 10.5 | 51.9 | 30.8 |
| Indian | Roti Canai 6.0<br>Chapatti 4.5 | Nasi Lemak 15.8<br>Kuih 0.3<br>Nasi Goreng 3.0<br>Noodle soup 8.3<br>Fried rice 3.8 | | | Bread 15.0<br>Biscuit, croissant 9.0<br>Sandwich 3.0<br>Cereal 3.8 |
| | 8.6 | 28.5 | 12.9 | 50.1 | 30.3 |
| Chinese | Noodle Soup 4.8<br>Fried Rice 3.8 | Nasi Lemak 10.6<br>Kuih 3.4<br>Nasi Goreng 0.2<br>Roti Canai 7.8<br>Chiapatti 4.6 | | | Bread 16.9<br>Biscuit, croissant 6.2<br>Sandwich 3.6<br>Cereal 3.6 |

The most focused on their own culinary style are Malays 18.3%, followed by Indians 10.5% and Chinese 8.6%. The most open to other typical styles are Indians 30.8%, Chinese 28.5% and Malays 20.5%. We can observe that the highest consumption rate for nasi lemak is among the Indian community and not the Malay one.

Indians and Chinese show the highest westernisation rates at 30.8% and 30.3%, respectively. Malays, meanwhile, are at 24.4%. However, it is this shared culinary background of the different communities that constitutes the common register of the Malaysian breakfast. In MFB, some questions have been asked at the request of the Ministry of Culture and Tourism, which intends to submit an emblematic Malaysian dish to UNESCO's intangible heritage list. The objective was to identify which dishes could be a good proposal from the population's point of view. It is interesting to note that at this level of identity representation, when asked, "Which dish could best represent Malaysia?", the population places "nasi lemak" far ahead, with more than 42.2%. Roti canai comes second

with 22.2%, followed by satays with 9.5% [37]. Therefore, nasi lemak and roti canai are two breakfast dishes that are considered to best represent the Malaysian food identity by Malaysians themselves. The place that the nutritional controversies (the consumption of Nasi Lemak on a regular basis was pointed to as a potential cause of overweight in Malaysia and sparked a debate. Some nutritionists were stressing the quantity of fat and calories—a regular portion of nasi lemak with a boiled egg, sambal and chicken accounts for approximately between 500 and 700 calories, 13 to 15 grams of protein, more than 14 grams of fat and more or less 80 grams of carbohydrates. Other voices, echoed by *TIME Magazine* in 2016, listed nasi lemak as one of ten most healthy international breakfasts. We can add that the question is difficult to arbitrate as the composition (and therefore the nutritional intake) varies considerably from one place of purchase to another) on nasi lemak has taken in media, far beyond Malaysia, is the sign of this identity dimension. Instead of blacklisting nasi lemak for its fat content, thus pointing negatively to a symbol of Malaysian identity, it might be in our interest to ask ourselves the question: "How can the nutritional profile of nasi lemak be improved?". However, this question opens some issues in the Malaysian context. Firstly, nasi lemak can be sourced from diverse types of stakeholders in the food system. To revise the nutritional profile, it would be necessary to work with the owners of restaurants and mamaks, as well as with all small traders involved in the more or less informal economy. How would it be possible to access these actors of the food system? And secondly, how would it be possible to ensure acceptance of the reprofiling of dishes with such emblematic meanings?

### 4.2. Organisation of the Food Day Patterns

The time distribution of meals during the day displayed in Figure 1 shows long time zones for all meals: from 6 to 10 h for breakfast (87.2%), from 11 to 15 h for lunch (93.9%) and from 18 to 23 h for dinner (dinner 93.8% and supper 21.7%). The time window of the mains meals is quite large and this temporary organisation could give the impression to the outside observer that Malaysians eat all the time.

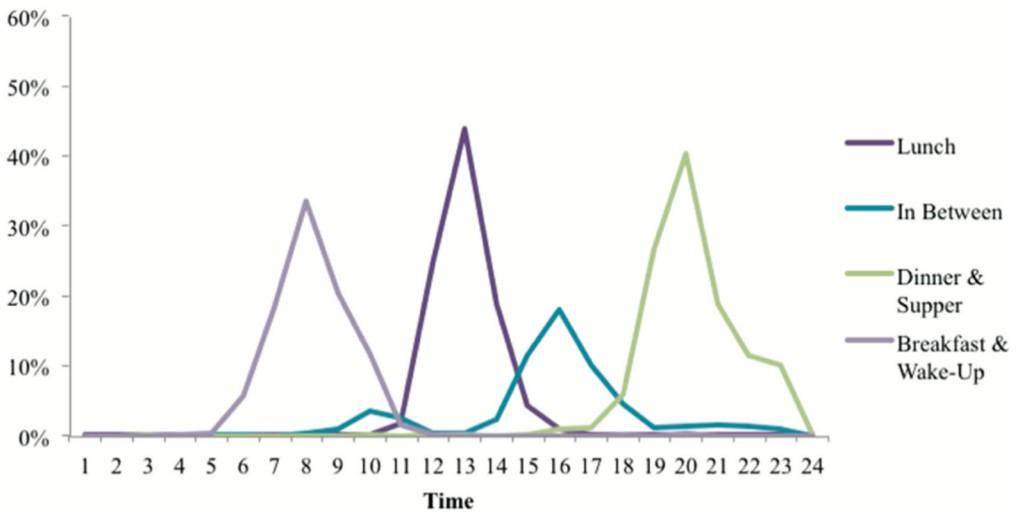

**Figure 1.** Food day temporal distribution.

As shown in Figure 2, seventy-six percent of the Malaysian sample have a food day organisation based on three meals a day (31% only the three meals and 45% three meals and other food intake(s)). Twenty-four percent of the interviewees have food days with two mains (15%, two meals and other intake(s), 7%, two meals only and 2%, one meal and other intake(s)). In this category, 52.3% do not eat breakfast. Half of these eat two meals that include a lunch and one meal in the evening. The other half of this category (47.7%) eat a morning meal and either a lunch or an evening meal. Non-Malay Bumiputras and less urbanised and modernised Malay people are over-represented in the two meals per

day category. Therefore, like in other parts of Asia Pacific [34], a coexistence between two main food day organisations can be observed.

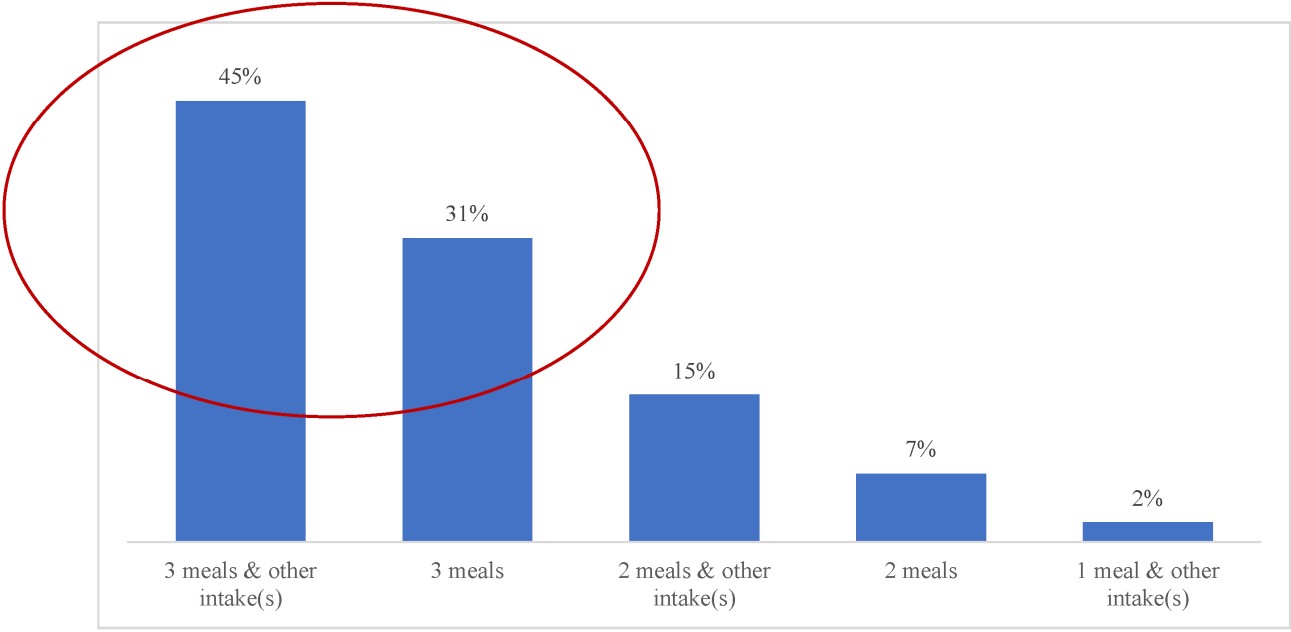

**Figure 2.** Organisation of the food days.

*4.3. Social Norms, Practices and BMI*

MFB makes it possible to analyse the gap between social norms and practices. Consequently, it is possible to expand the classical approach of nutritional patterns [38] by introducing cognitive discrepancy between social norms and practices [22]. Early analyses of eating patterns from MFB have showed that the prevalence of obesity is higher in the case of discrepancy [19].

The total share of people in dissonance is 23.64% for the global sample. It is 24.9% for Malay and non-Malay Bumiputra (NMB), 21% for Indians and only 13.9% for Chinese (Figure 3). These gaps could potentially be a cause of dissonance and a source of anxiety.

What is the relationship between the norms produced by nutritional sciences and disseminated by public health nutritionists and the social and cultural norms that govern food? This question cannot be reduced to a simple contradiction between scientifically based norms and social norms marked by a certain cultural relativism, and which should be reformed. The objective of this article is to draw the attention of the scientific community of public health nutrition to the possible consequences and risks of ineffectiveness of prescriptions that do not consider the social and cultural dimensions of food. This is the case, for example, of "counterproductive" effects that could be the consequence of food anxiety following the dissemination of messages that are difficult to apply because they contradict the social norms of the country and culture concerned. Hence, it is crucial to identify social norms that govern food practices and their social functions and to define how the dissemination of advances in nutritional science knowledge plays a part in redefining them.

In other words, what are the traditional conceptions of breakfast and how does modern nutrition challenge them or, on the contrary, on certain points, validate them? The objective here is not only to point out the contradictions, but also to see how they could be overcome. Since social standards relate to food and to practicalities of food preparation and consumption, while nutritional standards relate to nutrient intake and balance, there is potentially a space for articulation between them.

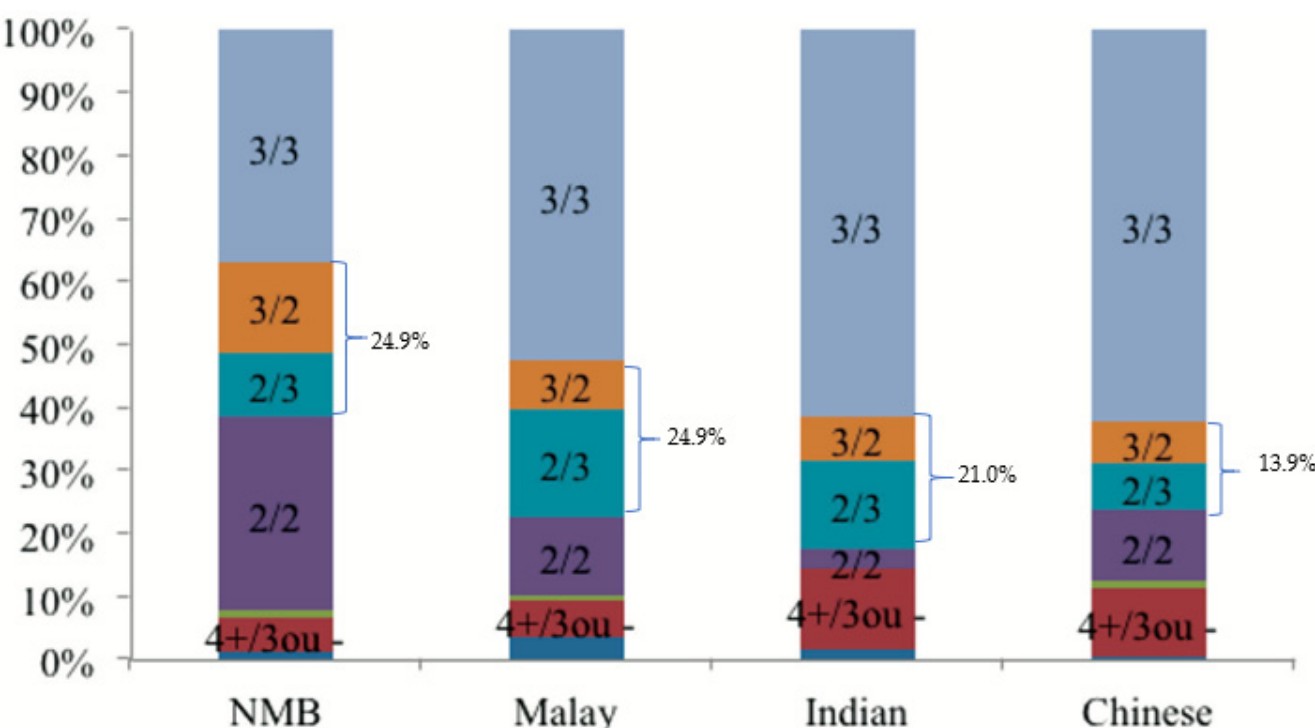

**Figure 3.** Comparison of the number of meals in norms and practices by ethnicity.

### 4.4. Breakfast as a Place of Reading Transformation of Food Habits and of Societies

How can we interpret what we have called "westernisation", either in the use of one or other of the Western breakfast structures (Continental or English) or the use of products such as cereals, biscuits, etc.? Several explanations are possible and can also be partly complementary. It may be the influence of British colonisation. From this point of view, the dominant use of the word breakfast, even in Malay, could be a sign pointing in this direction. The presence of a collation in the afternoon designated as "teatime" could be an additional sign.

However, it can also be the result of the marketing of breakfast products (cereals and biscuits), whose presence is notable on supermarket shelves, or even the offerings of certain restaurants. This is generally the cause chosen by nutritional transition theorists [38]. This type of explanation is also compatible with defenders of local food cultures. However, in the case of Malaysia, the reading grid of "compressed modernisation" [39] could turn out to be interesting and relevant.

Our position is that breakfast in Malaysia is a nested identity practice. It makes it possible to affirm multiple affiliations: the assigned affiliation group and the Malaysian community as a whole. The westernised breakfast could enter a cosmopolitan configuration. We find here, but with nuances, the forms of multiple affiliations brought to light by Laurence Tibère in La Reunion society [40].

The study of breakfast leads to the question of the organisation of the food day itself. How many meals do the individuals in the study population eat per day? Two, three, more than three? How should "other" food intakes be considered? And if we place ourselves on the normative level, how much should we consume?

The vulgate of modern nutrition promotes three meals a day and considers it, explicitly, as a "universal" practice that is, moreover, biologically based. However, this does not appear to be that obvious. Anthropologists, sociologists and historians have long pointed out the variability of the systems of social norms that influence meals and food days [22,34,41–44]. Variability occurs between cultures and within a culture over time. The phenomenon also concerns contemporary societies [45]. Furthermore, the arbitration of the number of meals per day on a strictly nutritional level is still the subject of debate.

In the conclusion of a literature review on the question, Bellisle, McDevitt and Prentice wrote: "at equivalent energy intakes, there are no scientific arguments to recommend one, two, three, four . . . food intakes" per day [46]. Research is still in progress to try to understand meal patterns [47], food day patterns [48] and their impact on nutrient intake and diet quality, as well as on body mass index [49]. Even the effects of the presence or absence of breakfast seems to be difficult to arbitrate [50], including when we introduce the circadian timing system [51].

How did this three-meal norm become almost universal? It is not only the result of the promotion of nutritional discourse. It also results from a particular organisation of working time; the forty-hour work week with a start of the working day between 8 a.m. and 9 a.m. and a meridian break and the end of the day between 5 p.m. and 7 p.m., which, from the 1960s, imposed itself as a model of "social progress" [22]. This temporal organisation was first adopted for office workers and then industries not using shift work. With economic transition, which is characterised by the expansion of the tertiary sector and the decline in agricultural employment, it concerns more and more people and has become the dominant model.

The promotion of the three meals a day model can be seen as one of the characteristics of "food modernisation". The new fact is that there is now only one standard: the three meals in the morning, noon and evening, and the (other) forms of food days are seen as deviations from the common norm rather than different norms [7]. This observation, already made several decades ago in the French context, applies perfectly to other cultural spaces such as Malaysia or Indonesia, which experienced later modernisation.

We see here how social norms and medical norms intertwine and how the social norms of scientists can slip into the health discourse and come to support it. In intercultural situations or in health promotion operations, it is important to be attentive to these stowaways that pass through the public health discourse.

## 5. Conclusions: Toward a Sociocultural Sustainability of Diets

Under the umbrella of Food and Agriculture Organization of the United Nations (FAO), a definition of Sustainable diets is proposed as "nutritionally adequate and healthy, safe, culturally acceptable, economically fair and affordable and having little environmental impact" [12]. However, the classic theories of sustainability are based on three dimensions: economic, social and environmental.

Sustainable diets focus on the impact of food choices on the health of the population and on the planet. These perspectives have introduced the social dimension of food choice alongside their economic and environmental dimensions. Food cultures are involved at two levels. First, they influence food choices and decisions, and second, they pre-determine food habits and practices by framing them in patterns, routines and scenarios of action.

The first level is the dominant perspective in public health nutrition. Social representations, believes and taboos are potentially taken into consideration as a determinant of choices. The influence of food cultures sometimes clashes with the nutritional recommendations and must be reformed using rational science-based explanations. Sometimes they converge and nutritional recommendations can leverage the cultural dimensions that must be reinforced.

A similar perspective is adopted in the understanding of the environmental impact of food choices. Therefore, the influence of food cultures is viewed through their consequences on health and on environment. An additional vision of sustainable nutrition consists of considering the influence of the food choices on the food system and its organisation, including the social organisation, employment, social hierarchy and inequality, gender equity [52,53], wellbeing, social cohesion, animal welfare and food sovereignty [16,54].

The second level, contribution of food sociology and anthropology, is to show that food choices are embodied in socio-cultural patterns and in social scenarios. Therefore, the levers of change in food habits are not only on the level of the rationality of the indi-

vidual but also at the level of the values system attached to the products, the dishes and the way to prepare and to consume them [26,55], in other words, at the level of the "food social space" [7]. There is a "proper way to do things related to food, that respects social order" [56].

Another contribution is to point out the fact that food social dimensions play a role in the organisation of the food system [57,58] and in the social system itself. Food choices have links with the construction, the expression and the management of social identities [59–61] and are involved in the construction of national identity [62–66]. Thus, the presence of nasi lemak and roti canai in the Malaysian breakfast supports technical–economic organisation and small businesses, stalls, mamaks, etc., and are, at the same time, the incarnation of the multicultural identity of Malaysia.

Several authors have highlighted the difficulty of considering ethnocultural factors related to food in the framework of the theories of sustainability [16]. To do this we have suggested distinguishing the social and ethno-cultural dimensions, differentiating issues related to inequalities in social hierarchy, gender, etc., and those related to ethno-cultural differences [67]. In the case of food, the first corresponds to issues of what we used to call "social sustainability" [53], while the second considers food cultures, types of food, the ways of preparing and consuming and the relations to social and cultural orders.

Within a few decades, Malaysia has experienced demographic and epidemiologic transition [68,69], that is to say the transition from mortality rates based on epidemic diseases, whose severity was reinforced by food scarcity, to a higher incidence of mortality through non-communicable diseases (NCDs) for which obesity is a significant risk factor [1,70]. The Malaysian Association of the Study of Obesity (MASO), founded in 1994, mobilises public and private actors to face the increase in obesity prevalence and other related medical problems that are now a major problem in the country [71,72]. The fight against the development of obesity, with the dissemination of nutritional knowledge, contributes to the individualisation of the relationship to food and more generally to the development of food anxiety.

When the "nutritionalisation" of modern societies puts eaters in a conflicting position between sociocultural norms and nutritional norms that can result in anxiety and social anomia, it can lead some people to eating disorders [5,22,27,73–78]. Therefore, it is important to articulate nutritional advice with the sociocultural dimensions of food. This attitude is expected to avoid counterproductive effects and thus could contribute to increasing the food wellbeing of the population and the efficiency of public health nutrition actions [79].

Ethno-cultural dimensions of food habits are now on the Malaysian research agenda [6,18,65,80–85]. This paper shows how knowledge of the ethno-cultural dimensions of breakfast could help public health nutritionists and policymakers deal with cultural characteristics and avoid the risk of a (non-conscious) neo-colonial attitude. The use of local dietary recommendations and a compositions table is a first stage to be able to consider local products and dishes. The second stage is to introduce a vision of local food cultures by considering ethno-cultural patterns and their social functions. Finally, this knowledge will contribute to debates around, and the progress towards, sociocultural sustainable healthy diets.

**Author Contributions:** Conceptualization, J.-P.P.; Methodology, J.-P.P.; Validation, J.-P.P., E.M., J.K., J.L.Y., L.T., C.L., F.-M.Y., A.D., P.K.N., N.A.R. and I.M.N.; Formal analysis, E.M.; Resources, J.-P.P.; Writing—original draft, J.-P.P.; Writing—review & editing, J.-P.P., E.M., J.K., J.L.Y., L.T., C.L., F.-M.Y., P.K.N., N.A.R. and I.M.N.; Funding acquisition, J.-P.P. All authors have read and agreed to the published version of the manuscript.

**Funding:** This research received indirect external funding from Cereal Partners Worldwide. (CPW-TU 2020) and from Centre National de la Recherche Scientifique, France (CNRS LIA-2020).

**Institutional Review Board Statement:** The funders had no role in the study design, data collection, analysis, interpretation, writing, decision to publish or preparation of the manuscript.

**Informed Consent Statement:** Not applicable.

**Data Availability Statement:** The primary database is available https://www.frontiersin.org/articles /10.3389/fnut.2021.800317/full (accessed on 28 January 2023).

**Acknowledgments:** The authors would like to acknowledge Mike Gibney and Sinead Hopkins.

**Conflicts of Interest:** J.P.P. has received grants for their respective institutions in generating data and honoraria from public agencies, non-profit organisations and private entities with an interest in socio-cultural dimensions of food patterns and the total diet.

## Appendix A  Interviewee Profiles

Interviewee 1 (expert): Malay

Age: 40+
Gender: Female
Profile: Senior lecturer and Head of Department—School of Liberal Arts and Sciences, in a private university
Doctor of Philosophy (Sociology and Anthropology), International Islamic University Malaysia, Malaysia

Interviewee 2 (expert): Malay

Age: 40+
Gender: Female
Profile: Senior lecturer and Head of School—School of Food Studies & Gastronomy in a private University,
MBA in Hospitality Administration, in US university.

Interviewee 3: Tamil

Age: 43
Gender: male
Lecturer in hospitality in a private Malaysian university
Master's degree in international hospitality management, Malaysian and European universities
Family language background: Tamil

Interviewee 4 (native speaker): Chinese (Mandarin and Cantonese)

Age: 27
Gender: Female
Education: Mandarin-based kindergarten, Primary school: SJK(C), secondary school: SMK.
Total exposure to mandarin class: 13 years.
Years speaking the language: 22 years for mandarin, 27 years for Cantonese
Family language background: Cantonese

Interviewee 5 (native speaker): Chinese (Mandarin)

Age: 42
Gender: Male
Years speaking the language: 42
Family language background: Mandarin

Interviewee 6 (native speaker): Chinese (Hokkien)

Age: 60
Gender: Female
Years speaking language: 60
Family background: Hokkien

Interviewee 7: Tamil

Age: 27
Gender: Female
Years speaking the language: 26 years
Family language background: Tamil

# Appendix B  Table of Data and Correlations

**Table A1.** Breakfast styles and Sociocultural characteristics.

| Socio-Demographic Variables | MFB1 | Breakfast Content | | | |
|---|---|---|---|---|---|
| | | Asian | Westernised | No Breakfast | Beverage Only |
| | N (%) | N (%) | N (%) | N (%) | N (%) |
| All | 1915 (100) | 1021 (53.3) | 535 (27.9) | 254 (13.3) | 105 (5.5) |
| Gender | | | | | |
| Male | 979 (51.1) | 516 (52.7) | 285 (29.1) | 125 (12.8) | 53 (5.4) |
| Female | 936 (48.9) | 505 (54.0) | 250 (26.7) | 129 (13.8) | 52 (5.6) |
| *p*-Value [1] | 0.679 | | | | |
| Urbanisation | | | | | |
| Urban | 1289 (67.3) | 680 (52.8) | 370 (28.7) | 175 (13.6) | 64 (5.0) |
| Suburban | 259 (13.5) | 129 (49.8) | 53 (20.5) | 60 (23.2) | 17 (6.6) |
| Rural | 367 (19.2) | 1021 (53.3) | 112 (30.5) | 254 (13.3) | 105 (5.5) |
| *p*-Value [1] | 0.000 * | | | | |
| Metropolisation | | | | | |
| Sabah Sarawak | 350 (18.3) | 210 (60.0) | 98 (28.0) | 25 (7.1) | 17 (4.9) |
| Rural Peninsular | 699 (36.5) | 254 (50.6) | 204 (29.2) | 111 (15.9) | 30 (4.3) |
| Metropolitan Areas | 866 (45.2) | 457 (52.8) | 233 (26.9) | 254 (13.3) | 58 (6.7) |
| *p*-Value [1] | 0.0001 * | | | | |
| Age | | | | | |
| 15–19 | 299 (15.6) | 164 (54.8) | 84 (28.1) | 38 (12.7) | 13 (4.3) |
| 20–29 | 544 (28.4) | 283 (52.0) | 151 (27.8) | 79 (14.5) | 31 (5.7) |
| 30–39 | 404 (21.1) | 219 (54.2) | 109 (27.0) | 56 (13.9) | 20 (5.0) |
| 40–49 | 330 (17.2) | 165 (50.0) | 98 (29.7) | 47 (14.2) | 20 (6.1) |
| 50–59 | 258 (13.5) | 141 (54.7) | 74 (28.7) | 28 (10.9) | 15 (5.8) |
| 60+ | 80 (4.2) | 49 (61.3) | 19 (23.8) | 6 (7.5) | 6 (7.5) |
| *p*-Value [1] | 0.870 | | | | |
| Ethnicity | | | | | |
| Non-Malay Bumiputra | 186 (9.7) | 110 (59.1) | 56 (30.1) | 11 (5.9) | 9 (4.8) |
| Malay | 1124 (58.7) | 593 (52.8) | 287 (25.5) | 192 (17.1) | 52 (4.6) |
| Indian | 126 (6.6) | 69 (54.8) | 41 (32.5) | 8 (6.3) | 8 (6.3) |
| Chinese | 479 (25.0) | 249 (52.0) | 151 (31.5) | 43 (9.0) | 36 (7.5) |
| *p*-Value [1] | 0.000 * | | | | |
| Occupation | | | | | |
| Active | 1501 (78.4) | 789 (52.6) | 419 (27.9) | 207 (13.8) | 86 (5.7) |
| Inactive | 414 (21.6) | 232 (56.0) | 116 (28.0) | 47 (11.4) | 19 (4.6) |
| *p*-Value [1] | 0.398 | | | | |
| Education Attainment | | | | | |
| Up to Primary School | 207 (10.8) | 125 (60.4) | 55 (26.6) | 14 (6.8) | 13 (6.3) |
| Lower Secondary School | 519 (27.1) | 293 (56.5) | 167 (32.2) | 38 (7.3) | 21 (4.0) |
| Upper Secondary School | 832 (43.4) | 407 (48.9) | 226 (27.2) | 150 (18.0) | 49 (5.9) |
| Higher Education | 357 (18.6) | 196 (54.9) | 87 (24.4) | 52 (14.6) | 22 (6.2) |
| *p*-Value [1] | 0.000 * | | | | |
| Monthly Available Income per Person (Ringit Malaysia) | | | | | |
| 100 to 699.99 | 474 (24.8) | 245 (51.7) | 140 (29.5) | 63 (13.3) | 26 (5.5) |

**Table A1.** *Cont.*

| Socio-Demographic Variables | MFB1 | Breakfast Content | | | |
|---|---|---|---|---|---|
| | | Asian | Westernised | No Breakfast | Beverage Only |
| | N (%) | N (%) | N (%) | N (%) | N (%) |
| 700 to 1332.99 | 705 (36.8) | 364 (51.6) | 185 (26.2) | 114 (16.2) | 42 (6.0) |
| 1333 to 1999.99 | 272 (14.2) | 151 (55.5) | 83 (30.5) | 26 (9.6) | 12 (4.4) |
| 2000 and above | 464 (24.2) | 261 (56.3) | 127 (27.4) | 51 (11.0) | 25 (5.4) |
| *p*-Value [1] | 0.144 | | | | |
| Number of family members staying together (including self) | | | | | |
| 1 pers | 33 (1.8) | 11 (32.4) | 5 (14.7) | 16 (47.1) | 2 (5.9) |
| 2–4 | 837 (43.7) | 451 (53.9) | 236 (28.2) | 97 (11.6) | 53 (6.3) |
| 5–6 | 783 (40.9) | 404 (51.6) | 222 (28.4) | 120 (15.3) | 37 (4.7) |
| 7–8 | 190 (9.9) | 114 (60.0) | 49 (25.8) | 16 (8.4) | 11 (5.8) |
| 9–10 | 52 (2.7) | 31 (59.6) | 15 (28.8) | 4 (7.7) | 2 (3.8) |
| More than 10 | 19 (1.0) | 10 (52.6) | 8 (42.1) | 1 (5.3) | 0 (0.0) |
| *p*-Value [1] | 0.000 * | | | | |
| Modernisation | | | | | |
| Low | 411 (21.5) | 245 (59.6) | 119 (29.0) | 25 (6.1) | 22 (5.4) |
| Medium | 921 (48.1) | 478 (51.9) | 255 (27.7) | 135 (14.7) | 53 (5.8) |
| High | 583 (30.4) | 298 (51.1) | 161 (27.6)) | 94 (16.1) | 30 (5.1) |
| *p*-Value [1] | 0.000 * | | | | |

[1] Chi-square test for association with $\alpha = 0.05$, * significant result.

**Table A2.** Breakfast, no breakfast.

| Socio-Demographic Variables | | Breakfast/No Breakfast | |
|---|---|---|---|
| | | No Breakfast | Breakfast |
| | N (%) | N (%) | N (%) |
| All | 2000 (100) | 251 (12.5) | 1749 (87.5) |
| Gender | | | |
| Male | 1016 (50.8) | 141 (13.9) | 875 (86.1) |
| Female | 984 (49.2) | 110 (11.2) | 874 (88.8) |
| *p*-Value [1] | 0.069 | | |
| Urbanisation | | | |
| Urban | 1351 (67.6) | 129 (9.5) | 1222 (90.5) |
| Suburban | 270 (13.5) | 60 (22.2) | 210 (77.8) |
| Rural | 379 (19.0) | 62 (16.4) | 317 (83.6) |
| *p*-Value [1] | 0.000 * | | |
| Metropolisation | | | |
| Sabah Sarawak | 362 (18.1) | 120 (33.1) | 242 (66.9) |
| Rural Peninsular | 730 (36.5) | 72 (9.9) | 658 (90.1) |
| Metropolitan Areas | 908 (45.4) | 59 (6.5) | 849 (93.5) |
| *p*-Value [1] | 0.000 * | | |
| Age | | | |
| 15–19 | 309 (15.5) | 51 (16.5) | 258 (83.5) |

**Table A2.** *Cont.*

| Socio-Demographic Variables | | Breakfast/No Breakfast | | |
| --- | --- | --- | --- | --- |
| | | | No Breakfast | Breakfast |
| | | N (%) | N (%) | N (%) |
| 20–29 | | 578 (28.9) | 69 (11.9) | 509 (88.1) |
| 30–39 | | 419 (21.0) | 50 (11.9) | 369 (88.1) |
| 40–49 | | 346 (17.3) | 51 (14.7) | 295 (85.3) |
| 50–59 | | 266 (13.3) | 25 (9.4) | 241 (90.6) |
| 60+ | | 82 (4.1) | 5 (6.1) | 77 (93.9) |
| | *p*-Value [1] | 0.038 | | |
| Ethnicity | | | | |
| Non-Malay Bumiputra | | 193 (9.7) | 69 (35.8) | 124 (64.2) |
| Malay | | 1176 (58.8) | 126 (10.7) | 1050 (89.3) |
| Indian | | 133 (6.7) | 6 (4.5) | 127 (95.5) |
| Chinese | | 498 (24.9) | 50 (10.0) | 448 (90.0) |
| | *p*-Value [1] | 0.000 * | | |
| Occupation | | | | |
| Active | | 1566 (78.3) | 208 (13.3) | 1358 (86.7) |
| Inactive | | 434 (21.7) | 43 (9.9) | 391 (90.1) |
| | *p*-Value [1] | 0.060 | | |
| Education Attainment | | | | |
| Up to Primary School | | 215 (10.8) | 31 (14.4) | 184 (85.6) |
| Lower Secondary School | | 543 (27.2) | 84 (15.5) | 459 (84.5) |
| Upper Secondary School | | 869 (43.5) | 89 (10.2) | 780 (89.8) |
| Higher Education | | 373 (18.7) | 47 (12.6) | 326 (87.4) |
| | *p*-Value [1] | 0.028 | | |
| Monthly Available Income per Person (Ringit Malaysia) | | | | |
| 100 to 699.99 | | 495 (24.8) | 60 (12.1) | 435 (87.9) |
| 700 to 1332.99 | | 739 (37.0) | 90 (12.2) | 649 (87.8) |
| 1333 to 1999.99 | | 281 (14.1) | 40 (14.2) | 241 (85.8) |
| 2000 and above | | 485 (24.3) | 61 (12.6) | 424 (87.4) |
| | *p*-Value [1] | 0.825 | | |
| Number of family members staying together (including self) | | | | |
| 1 pers | | 34 (1.7) | 6 (17.6) | 28 (82.4) |
| 2–4 | | 875 (43.8) | 104 (11.9) | 771 (88.1) |
| 5–6 | | 815 (40.8) | 119 (14.6) | 696 (85.4) |
| 7–8 | | 200 (10.0) | 14 (7.0) | 186 (93.0) |
| 9–10 | | 54 (2.7) | 5 (9.3) | 49 (90.7) |
| More than 10 | | 22 (1.1) | 3 (13.6) | 19 (86.4) |
| | *p*-Value [1] | 0.063 | | |
| Modernisation | | | | |
| Low | | 425 (21.3) | 72 (16.9) | 353 (83.1) |
| Medium | | 968 (48.4) | 116 (12.0) | 852 (88.0) |
| High | | 607 (30.4) | 63 (10.4) | 544 (89.6) |
| | *p*-Value [1] | 0.006 * | | |

[1] Chi-square test for association with $\alpha = 0.05$, * significant result.

**Table A3.** Number of meals a day and social characteristics.

| Socio-Demographic Variables | MFB1 | Number of Meals per Day | |
|---|---|---|---|
| | | 2 Meals | 3 Meals |
| | N (%) | N (%) | N (%) |
| All | 1970 (100) | 447 (22.7) | 1523 (77.3) |
| Gender | | | |
| Male | 998 (50.7) | 225 (22.5) | 773 (77.5) |
| Female | 972 (49.3) | 222 (22.8) | 750 (77.2) |
| *p*-Value [1] | 0.0.876 | | |
| Urbanisation | | | |
| Urban | 1337 (67.9) | 261 (19.5) | 1076 (80.5) |
| Suburban | 265 (13.5) | 95 (35.8) | 170 (64.2) |
| Rural | 368 (18.7) | 91 (24.7) | 277 (75.3) |
| *p*-Value [1] | 0.000 * | | |
| Metropolisation | | | |
| Sabah Sarawak | 355 (18.0) | 160 (45.1) | 195 (54.9) |
| Rural Peninsular | 713 (36.2) | 139 (19.5) | 574 (80.5) |
| Metropolitan Areas | 902 (45.8) | 148 (16.4) | 754 (83.6) |
| *p*-Value [1] | 0.000 * | | |
| Age | | | |
| 15–19 | 304 (15.4) | 82 (27.0) | 222 (73.0) |
| 20–29 | 559 (28.9) | 132 (23.2) | 437 (76.8) |
| 30–39 | 411 (20.9) | 86 (20.9) | 325 (79.1) |
| 40–49 | 343 (17.4) | 86 (25.1) | 257 (74.9) |
| 50–59 | 262 (13.3) | 50 (19.1) | 212 (80.9) |
| 60+ | 81 (4.1) | 11 (13.6) | 70 (86.4) |
| *p*-Value [1] | 0.054 | | |
| Ethnicity | | | |
| Non-Malay Bumiputra | 191 (9.7) | 88 (46.1) | 103 (53.9) |
| Malay | 1157 (58.7) | 257 (22.2) | 900 (77.8) |
| Indian | 131 (6.6) | 11 (8.4) | 120 (91.6) |
| Chinese | 491 (24.9) | 91 (18.5) | 400 (81.5) |
| *p*-Value [1] | 0.000 * | | |
| Occupation | | | |
| Active | 1542 (78.3) | 356 (23.1) | 1186 (76.9) |
| Inactive | 428 (21.7) | 91 (21.3) | 337 (78.7) |
| *p*-Value [1] | 0.425 | | |
| Education Attainment | | | |
| Up to Primary School or lower | 210 (10.7) | 56 (26.7) | 154 (73.3) |
| Lower Secondary School | 531 (27.0) | 131 (24.7) | 400 (75.3) |
| Upper Secondary School | 858 (43.6) | 175 (20.4) | 683 (79.6) |
| Higher Education | 371 (18.8) | 85 (22.9) | 286 (77.1) |

**Table A3.** *Cont.*

| Socio-Demographic Variables | MFB1 | Number of Meals per Day | |
|---|---|---|---|
| | | 2 Meals | 3 Meals |
| | N (%) | N (%) | N (%) |
| *p*-Value [1] | 0.129 | | |
| Monthly Available Income per Person (Ringit Malaysia) | | | |
| 100 to 699.99 | 488 (24.8) | 110 (22.5) | 378 (77.5) |
| 700 to 1332.99 | 729 (37.0) | 171 (23.5) | 558 (76.5) |
| 1333 to 1999.99 | 275 (14.0) | 63 (22.9) | 212 (77.1) |
| 2000 and above | 478 (24.3) | 103 (21.5) | 375 (78.5) |
| *p*-Value [1] | 0.893 | | |
| Number of family members staying together (including self) | | | |
| 1 pers | 33 (1.7) | 15 (45.5) | 18 (54.5) |
| 2–4 | 859 (43.6) | 189 (22.0) | 670 (78.0) |
| 5–6 | 806 (40.9) | 188 (23.3) | 618 (76.7) |
| 7–8 | 197 (10.0) | 35 (17.8) | 162 (82.2) |
| 9–10 | 53 (2.7) | 11 (20.8) | 42 (79.2) |
| More than 10 | 22 (1.1) | 9 (40.9) | 13 (59.1) |
| *p*-Value [1] | 0.004 * | | |
| Modernisation | | | |
| Low | 414 (21.0) | 109 (26.3) | 305 (73.7) |
| Medium | 956 (48.5) | 208 (21.8) | 748 (78.2) |
| High | 600 (30.5) | 130 (21.7) | 470 (78.3) |
| *p*-Value [1] | 0.138 | | |

[1] Chi-square test for association with $\alpha = 0.05$, * significant result.

**Table A4.** Breakfast at home/out of home.

| Socio-Demographic Variables | MFB1 | Breakfast At Home/Out of Home | |
|---|---|---|---|
| | | At Home | Out of Home |
| | N (%) | N (%) | N (%) |
| All | 1749 (100) | 976 (55.8) | 773 (44.2) |
| **Gender** | | | |
| Male | 875 (50.0) | 392 (44.8) | 483 (55.2) |
| Female | 874 (50.0) | 584 (66.8) | 290 (33.2) |
| *p*-Value [1]  0.000 * | | | |
| **Urbanisation** | | | |
| Urban | 1222 (69.9) | 670 (54.8) | 552 (45.2) |
| Suburban | 210 (12.0) | 124 (59.0) | 86 (41.0) |
| Rural | 317 (18.1) | 182 (57.4) | 135 (42.6) |
| *p*-Value [1]  0.427 | | | |
| **Metropolisation** | | | |
| Sabah Sarawak | 242 (13.8) | 186 (76.9) | 56 (23.1) |
| Rural Peninsular | 658 (37.6) | 332 (50.5) | 326 (49.5) |
| Metropolitan Areas | 849 (48.5) | 458 (53.9) | 391 (46.1) |
| *p*-Value [1]  0.000 * | | | |
| **Age** | | | |
| 15–19 | 258 (14.8) | 162 (62.8) | 96 (37.2) |
| 20–29 | 509 (29.1) | 251 (49.3) | 258 (50.7)) |
| 30–39 | 369 (21.1) | 188 (50.9) | 181 (49.1) |
| 40–49 | 295 (16.9) | 157 (53.2) | 138 (46.8) |
| 50–59 | 241 (13.8) | 162 (67.2)) | 79 (32.8) |
| 60+ | 77 (4.4) | 56 (72.7) | 21 (27.3) |
| *p*-Value [1]  0.000 * | | | |
| **Ethnicity** | | | |
| Non-Malay Bumiputra | 124 (7.1) | 107 (86.3) | 17 (13.7) |
| Malay | 1050 (60.0) | 591 (56.3) | 459 (43.7) |
| Indian | 127 (7.3) | 71 (55.9) | 56 (44.1) |
| Chinese | 448 (25.6) | 207 (46.2) | 241 (53.8) |
| *p*-Value [1]  0.000 * | | | |
| **Occupation** | | | |
| Active | 1358 (77.6) | 698 (51.4) | 660 (48.6) |
| Inactive | 391 (22.4) | 278 (71.1) | 113 (28.9) |
| *p*-Value [1]  0.000 * | | | |
| **Education Attainment** | | | |
| Up to Primary School | 184 (10.5) | 120 (65.2) | 64 (34.8) |
| Lower Secondary School | 459 (26.2) | 278 (60.6) | 181 (39.4) |
| Upper Secondary School | 780 (44.6) | 430 (55.1) | 350 (44.9) |
| Higher Education | 326 (18.6) | 148 (45.4) | 178 (54.6) |

<div align="center">Table A4. <em>Cont.</em></div>

| Socio-Demographic Variables | MFB1 | Breakfast At Home/Out of Home | |
|---|---|---|---|
| | | At Home | Out of Home |
| | N (%) | N (%) | N (%) |
| *p*-Value [1] | 0.000 * | | |
| Monthly Available Income per Person (Ringit Malaysia) | | | |
| 100 to 699.99 | 435 (24.9) | 262 (60.2) | 173 (39.8) |
| 700 to 1332.99 | 649 (37.1) | 334 (51.5) | 315 (48.5) |
| 1333 to 1999.99 | 241 (13.8) | 144 (59.8) | 97 (40.2) |
| 2000 and above | 424 (24.2) | 236 (55.7) | 188 (44.3) |
| *p*-Value [1] | 0.019 | | |
| Number of family members staying together (including self) | | | |
| 1 pers | 28 (1.6) | 9 (32.1) | 19 (67.9) |
| 2–4 | 771 (44.1) | 406 (52.7) | 365 (47.3) |
| 5–6 | 696 (39.8) | 391 (56.2)) | 305 (43.8) |
| 7–8 | 186 (10.6) | 123 (66.1) | 63 (33.9) |
| 9–10 | 49 (2.8) | 35 (71.4) | 14 (28.6) |
| More than 10 | 19 (1.1) | 12 (63.2) | 7 (36.8) |
| *p*-Value [1] | 0.000 * | | |
| Modernisation | | | |
| Low | 353 (20.2) | 217 (61.5) | 136 (38.5) |
| Medium | 852 (48.7) | 530 (62.2) | 322 (37.8) |
| High | 544 (31.1) | 229 (42.1) | 315 (57.9) |
| *p*-Value [1] | 0.000 * | | |

<div align="center"><sup>1</sup> Chi-square test for association with α = 0.05, * significant result.</div>

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
