# Peer review of "Much More Than Food: The Malaysian Breakfast, a Socio-Cultural Perspective"

_sustainability, doi:10.3390/su15032815_

Round 1

Reviewer 1 Report

The article is not highly original, but presents interesting data about food changes and food habits in a multicultural society. It is fairly well written, the style just needs to be fully checked. It seems that some words are not totally accurate.

The main problem, that can be easily corrected, is that the question of sustainability only comes in the conclusion. It splits the article in two. The whole article until the conclusion deals with the ethnocultural dimensions and social functions of breakfast in Malaysia, as well as food changes reflected in the breakfast. The conclusion brings up different arguments about social sustainability and public policies. These questions should be introduced since the beginning, to be argued at the end of the article. This would give more homogeneity to the article.

The different dishes cited in the article should be shortly described. A reader not specialized in the described region has no idea of what are nasi lemak, roti canai, congee, and so on. The term Bumiputra should also be explained. I suppose it refers to Orang Asli and native inhabitants of Sabah and Sarawak. All these last ethnic groups are put in the same category, but do Orang Asli eat the same type of breakfast as Borneo inhabitants? Their names for breakfast are not cited either, and they don't seem to be included in some of the results, such as table 3 (line 233) and following lines.

The paragraph on English and French breakfast (lines 102 to 121) could be reduced.

Author Response

Questions, remarks, and suggestions

Answers from authors

The article is not highly original but presents interesting data about food changes and food habits in a multicultural society. It is fairly well written; the style just needs to be fully checked. It seems that some words are not totally accurate. 

The article has been edited by a Professional editor and in the team some authors are native speakers. The expression cognitive dissonance has been changed for discrepancy (lines 289 and 291).

The main problem, that can be easily corrected, is that the question of sustainability only comes in the conclusion. It splits the article in two. The whole article until the conclusion deals with the ethnocultural dimensions and social functions of breakfast in Malaysia, as well as food changes reflected in the breakfast. The conclusion brings up different arguments about social sustainability and public policies. These questions should be introduced since the beginning, to be argued at the end of the article. This would give more homogeneity to the article.

The introduction has been rewritten to consider the suggestion. The added statement reads as “Finally, the multicultural Malaysian society will be a suitable empirical field to study in detail the distinction between social and cultural dimensions of food and to feed the debate on the conception of a sociocultural sustainable healthy diets [13,15,16].” (please refer to lines 74-77).

The different dishes cited in the article should be shortly described. A reader not specialized in the described region has no idea of what are nasi lemak, roti canai, congee, and so on.

Thank you for this suggestion. A footnote (number 3) has been inserted to provide short descriptions of the mentioned dishes and reads as “

"nasi lemak",

"nasi goreng",

"kuih";

"roti canai",

"chappati";

“dim sum”

Nasi lemak is a dish composed of rice cooked in coco milk, served with….

Nasi goreng is…  

Roti chanai is….

Note number 3

The term Bumiputra should also be explained. I suppose it refers to Orang Asli and native inhabitants of Sabah and Sarawak. All these last ethnic groups are put in the same category, but do Orang Asli eat the same type of breakfast as Borneo inhabitants?

A note (number 1) has been inserted at the footage of page 2. It reads as “The term non-Malay Bumiputra ("sons of the soil") refers to the indigenous inhabitants of Malaysia. It officially includes the Orang Asli (aboriginal populations of the peninsula) and the natives of Sabah and Sarawak.” Given the requirements for statistical analysis, the authors have chosen not to investigate the differences amongst non-Malay Bumiputra groups in the context of this article.

Their names for breakfast are not cited either, and they don't seem to be included in some of the results, such as table 3 (line 233) and following lines.

Thank you for highlighting this formatting issue. The title of the table 3 (page 6) has been revised accordingly and now reads as “Breakfast food and ethnicity, in Malay, Indian and Chinese Malaysians”

The paragraph on English and French breakfast (lines 102 to 121) could be reduced.

The paragraph (lines 105 to 122) is reduced from 287 words to 256 and now reads as “Let us take the example of an individual who "decides" to go for breakfast. Depending on the food culture they belong to, their reasoning and the choices they must make will differ widely. In some contexts, they will not even decide in the sense of a 'reflective act'. Take, for instance, the case of a French and a British person. The room of decision will first and foremost be within structures like “hot drink” + “bread or pastry” + “butter, jam/spread”. The choice will therefore be made within the structure of the hot drink; the bread, toast, biscuits, pastries; the butter and jam/spread. Other breakfast models are possible in this culture but not dominant. Variants can include more foods such as cheese, cold cuts, cereals, fruit juices... They result of several phenomena, first nutritional discourse in favour of a “hearty” breakfast, to which the food industry often follows suit, when it does not precede it. Second the aggregation of the French breakfast with the "case croute". This "small meal" taken by people who start to work early in the morning to complement a rather light breakfast. This combination of two morning intakes undoubtedly explains the large proportion of very simplified breakfasts (± 20%) observed in the French population [25]. The British person’s choices will be made within the framework of a more complex structure which includes eggs, beans, sausages, mushrooms, tomatoes, toast, etc. These “breakfast structures” are visible on the breakfast menus of international hotels, under the names "continental breakfast", "Full English breakfast" and the like.”

Reviewer 2 Report

This article is excellent overall in terms of both form and content and its relevance to the field of food studies. Moreover, the question of the first meal of the day (morning meal) is rather rarely treated and reflected in food studies compared to the other meals of the day ans is of great interest. I recommend accepting it as is.

Coming from the human and social sciences and food studies fields, I am always surprised at how much public health authorities (in this case, nutritionists) need to be reminded of the importance of social and cultural factors in the problematization and public policies on the life habits of populations. The authors rightly remind us that if the eating habits of the population are part of life habits, the latter (and more broadly foods), have a fundamental anthropological importance that goes far beyond the simple food intake in order to ensure the maintenance of the organism. Indeed, foods are constitutive of eaters’ and communities’ identity. Moreover, the authors show, and I am again surprised we still have to remind but it seems necessary, that the environments (here the Food Scapes or Food Cultures) are determining in the food choices that individuals make, because they mark out the field of possibilities for those “choices” to occur. Moreover, among other things, because food and its preparation are closely linked to emotions, to strong social constructs and to a multitude of (often contradictory) discourses, individuals' choices are only partly based on a rational planning of action.

The authors also show that a tool for measuring food practices, such as the Malaysian Food Barometer (MFB), is relevant in that it does not propose categories that are all already formatted (meals, foods, culinarity, etc.), but leaves it up to the eater to note down everything he or she eats on a daily basis and to group the items into "meals" or not, thus avoiding the bias of a preconception that would not take into account the social and cultural constructs of these categories for the Malaysian populations. The MFB results are therefore relevant to a cultural and social understanding of eating habits. Finally, the authors warn us in a relevant way against the reduction of food cultures to static entities and show well how it is necessary to take into account the processes of borrowing, cosmopolitanism, hybridization and, more broadly, the circulation of food and culinary techniques in social food spaces that are increasingly transcultural. The article clearly shows that this is the case at least in Malaysia's large cities.

The authors show that this sociocultural approach reverses (or shifts) the usual questioning of public health nutritionists. For example, the question is not how to reduce the consumption of Nasi Lemak at breakfast, but rather how to improve the composition of Nasi Lemak so that it has a "healthier" profile for daily consumption. And I do think that is relevant and promising from a public health perspective.

I would like to make two suggestions or comments for this paper that are minor.

The first is the use of the concept of cognitive dissonance. At the point where this concept makes its appearance in the text, it is not yet relevant in my opinion to present it, as there is a lack of a specific explanation for this phenomenon that will come later in the text (line 402 to 410) and where it would then be very relevant to bring in this concept, which is not otherwise there. I believe that according to the argument and the content presented, one cannot yet speak of cognitive dissonance, but rather of discrepancy or, at worst, of clash.

The second is the reversal of the question about nasi lemak. The authors show that the question should not be “how to reduce the consumption of Nasi Lemak at breakfast”, but rather “how to improve the composition of Nasi Lemak so that it has a more "healthy" profile”. What is a pity here is that the reflection stops here, when it opens very interesting perspectives on which it would have been very interesting for the authors to elaborate or discuss a little more. I am thinking, among other things, of the fact that it seems that a good part of the population that eats nasi lemak for breakfast does so outside their homes because this dish needs to be cooked and, for various reasons, it is more convenient to take it on the way to work or school. Thus, I understand that if work were to be done to "improve" the composition of this dish, it would be necessary to work with street food vendors to do so. Thus, there would be at least two huge challenges to overcome here, which are how to motivate these restaurant owners to make changes to the preparation of the dish and (this is closely related), how to make changes to an emblematic dish that would have cultural acceptability?

Author Response

Questions, remarks, and suggestions

Answers from authors

The first is the use of the concept of cognitive dissonance. At the point where this concept makes its appearance in the text, it is not yet relevant in my opinion to present it, as there is a lack of a specific explanation for this phenomenon that will come later in the text (line 402 to 410) and where it would then be very relevant to bring in this concept, which is not otherwise there. I believe that according to the argument and the content presented, one cannot yet speak of cognitive dissonance, but rather of discrepancy or, at worst, of clash.

The presentation of the conception of cognitive dissonance is too lengthy in such a paper. Thus, following the suggestion of the reviewer, we have decided to use the word discrepancy. The section now reads as follows (lines 287-291): “MFB makes it possible to analyse the gap between social norms and practices. Consequently, it is possible to expand the classical approach of nutritional patterns [38], by introducing cognitive discrepancy between social norms and practices [22]. Early analyses on eating patterns from MFB have showed that prevalence of obesity is higher in case of discrepancy [19].”

The second is the reversal of the question about nasi lemak. The authors show that the question should not be “how to reduce the consumption of Nasi Lemak at breakfast”, but rather “how to improve the composition of Nasi Lemak so that it has a more "healthy" profile”. What is a pity here is that the reflection stops here, when it opens very interesting perspectives on which it would have been very interesting for the authors to elaborate or discuss a little more. I am thinking, among other things, of the fact that it seems that a good part of the population that eats nasi lemak for breakfast does so outside their homes because this dish needs to be cooked and, for various reasons, it is more convenient to take it on the way to work or school. Thus, I understand that if work were to be done to "improve" the composition of this dish, it would be necessary to work with street food vendors to do so. Thus, there would be at least two huge challenges to overcome here, which are how to motivate these restaurant owners to make changes to the preparation of the dish and (this is closely related), how to make changes to an emblematic dish that would have cultural acceptability?

Thank you for your interest. A section has been added (lines 261-266) and reads as “However, this question opens some issues in the Malaysian context. Firstly, nasi lemak can be sourced from diverse types of stakeholders in the food system. To revise the nutritional profile, it will be mandatory to work with the owners of restaurants and mamaks, as well as with all small traders involved in the more or less informal economy. How to access these actors of the food system? And secondly, how to ensure acceptance of the reprofiling of dishes with such emblematic meanings?”

Reviewer 3 Report

The article deals with Malaysian breakfast from a socio-cultural perspective. It highlights, very convincingly, the tensions between the expression of cultural identities and nutritional recommendations in a situation of accelerated food modernization.

The point of view adopted aims to consider that eating behavior is not just a result of conscious decisions or relevant information. It stems from patterns and routines that are socially and culturally pre-defined. In this context, breakfast plays a special role in the organization of meals throughout the day.

The case studied shows a beautiful complexity, with practices with a strong cultural basis, according to the different ethnic groups present, in combination with various influences, including the Western continental breakfast. The results obtained are clearly set out and allow a real understanding of the situation. They show to what extent breakfast constitutes a "nested identity practice". They open up elements of a more general discussion on the composite nature of identities (multiple affiliations), on the relationship between nutrition science and public health prescriptions, and on their confrontation with social and cultural norms.

I have some major remarks proceeding from criticism I make (part A), then I formulate some suggestions for improving the reader comprehension (part B).

A – Major remarks: Some constructive criticism can be made.

1 - The Nasi Lemak, a central product in the demonstration.

Nasi Lemak seems to play a central role in this work because it is cited as early as L125. However, it is not really presented in its specificities and that is a pity. Possible variations in its preparation (ingredients, cooking methods) and their consequences on its nutrient composition should be explained. In addition, the text points out (L248) a controversy which, on reading, sheds light on the subject, but unfortunately, nothing more is said about this controversy. It is a major element of the whole article: the authors should introduce it correctly (and not stealthily in the middle of the results on the Malaysian style) and inform us about it. Who wears it, how it is expressed, is the fat content of Nasi Lemak essential for authenticity and taste, is its consumption correlated with the appearance of cases of obesity and type II diabetes, etc.? I recommend focusing on this emblematic food since it crystallizes the tensions studied.

In addition, the question posed (in rhetorical form) in L250-251 on fat and the possibilities of reducing it to improve its nutritional profile, is formulated without treatment modality. Attributed to the Malay community ("our interest"), this question seems to be a major result of the article. However, it is not developed at all in the rest of the article and the authors do not return to this point in the discussion and in the conclusion. This is a pity because it would give real grips on how the debates (desired in the last sentence of the article) could be fueled. I recommend making it an axis of the final discussion by giving ways of dealing with how to overcome the contradictions identified.

2 – Dissonance and anxiety, link with obesity: fragile affirmations.

The text reveals in L282 the sources of anxiety related to the “dissonances” between the declarative and the actual practices. Figure 3 is supposed to provide the reader with something to identify the proportions of individuals concerned in each ethnic group. It remains difficult to understand what this is all about. I had to consult the cited reference (Fournier et al, 2016, reference 19) to better understand what was presented. Do the authors really need these clarifications in the article when the readers need other elements to better understand the point?

In addition, this reference 19 tells us that cases of elevated BMI are not correlated with dissonance (“obesity is over-represented in people who have dissonant eating behaviors”) and that, therefore, this direction is not necessarily the one to be explored. Social norms and practices are often at odds, but it does not seem obvious to make them a causality of obesity cases.

There remains “anxiety”, which, according to the authors, is an important factor to consider. This remains a possibility, however not demonstrated by the data collected, and which would require open interviews with the persons concerned. It seems to me that this should be expressed with caution in absence of reliable data.

A relevant link is made at the end of the article (L413) with the risks of anomia and the need to articulate nutritional recommendations and socio-cultural practices. However, it does not require, in my opinion, to go through the individualized aspects of anxiety that are not at the heart of the article.

3 – Neo-colonialism and nutrition science, the role of public authorities.

In L288, the analysis switches to the link between nutritional science contributions and public health prescriptions, highlighting the risk of inefficiency of these prescriptions when they do not take into account social and cultural norms. This idea seems consistent with the above, even if it is not supported by any concrete results. In L290-291, in the rest of the reasoning, the authors identify messages impossible to respect because of contradictions with social norms. An illustration of such messages would be welcome at this stage, in order to give these possible contradictions a minimum of reality. For example, I suggest that a message that is very difficult to apply and regarding Nasi Lemak could be included in the article. If possible, related to the question about improving its nutritional profile.

In addition, nutritional sciences provide knowledge and not directly prescriptions and this knowledge is rarely communicated to the general public (because of its complexity and the levels of education it implies). The question (L292-295) becomes: how to integrate nutrition knowledge into the evolution of social and cultural norms that govern food practices without them becoming counterproductive? This point should distinguish between the production of knowledge by nutrition sciences and the enactment of prescriptions by public authorities which are supposed to be based on that knowledge, but in fact proceed from their interpretation. It is therefore not only questions of dissemination of knowledge (L294) but rather of their translation into prescription that should be addressed.

This distinction leads to another criticism when, in conclusion, sustainable diets aim to take into account social and cultural norms. The clash between food culture and nutritional recommendations (L372) determines a framework of frontal opposition, a kind of balance of power between the authorities and the people. From this confrontation, appears the risk of neo-colonialism (L420) which is not explained and only posed as a foil. This point deserves to be discussed more in-depth, through two proposed reflections:

i)                 it should not be science and knowledge that, per se, feed neo-colonialism but the way in which the authorities mobilize them in order to legitimize their recommendations and norms;

ii)                social and cultural norms are not fixed, they are constantly evolving and the major challenge becomes their ability to integrate nutritional knowledge for the benefit of populations by avoiding their de-legitimization and a form of acculturation.

The role of scientists themselves is just suggested in L354-355, with a supposed influence of their own social norms on their activity, and consequently, on the knowledge they produce. This point is rather delicate to deal with and concrete examples should be brought to this stowaway. Moreover, it seems to be a general question in all societies, with the presence, in the media and in societal debates, of some scientists who think they provide the truth able to organize the society. This point (and that of neo-colonialism that appears below) would make Malay scientists a Trojan horse of the Western powers on Malay society. Is this what the authors want readers to understand?

The final opening on local recommendations and composition charts seems very relevant to me and goes well in the direction of a re-legitimization of the traditional breakfast. However, here again, it would be good to link future debates to objects such Nasi Lemak: are variations in preparation and composition likely to enlighten the  buyer when conscious of nutritional recommendations? Could be possible to imagine a "light" Nasi Lemak without losing its ethnic character (its "typicity")? How should it be identified to distinguish it from the one that is too fat?

B – Minor remarks: Some suggestions to improve reader comprehension

1 – Ethnicities and languages

The level of expertise of readers on the situation in Malaysia is much lower than that of authors. Some additional information should be provided to help them in their understanding.

For example, Hokkien arrives in L188 without any mention of it above. Perhaps introducing the different dialects of the Chinese community before drawing up Table 1 would be useful.

Similarly, non-Malay Bumiputras in L269 are not defined above in the text. The whole paragraph about them becomes difficult to understand.

Finally, the word "satays" in L246 is absent from all the text and tables, so it is impossible for the average reader (including me) to understand.

2 – Breakfast and number of meals

The specificities of breakfast among daily meals are well presented. But as, in families, all members do not have the same schedules, this meal is often devoid of social ties, everyone taking his breakfast on his side. In addition, the practice of buying and "take-away" (the eater does not control the preparation or the ingredients) ends up reducing the scope of this meal in terms of social ties (a meal outside the family setting). It would be appropriate to point this out (L138).

The British breakfast includes cooked dishes unlike the Continental which does not. It is hard to think (L306) that British colonization could have influenced the Malay to choose continental breakfast as an alternative to traditional preparations that also include cooked products (L211). The term "westernise style" is relatively vague and it is only clear from Table 3 that it is continental and not British.

On a day-long scale, the morning meal plays a specific role in a large number of societies because it is necessary to get to work quickly and to be able to have sufficient reserves to be productive. However, the level of physical activity has decreased significantly between traditional society and current society. This point deserves to be mentioned insofar as nutritional recommendations must integrate energy needs according to the professions. In a peasant society with field work, breakfast must provide the necessary calories to the worker. It is therefore not surprising that food issued from this traditional society shows a relative inadequacy to current working conditions. This is an element of the analysis that should be introduced, especially since a nutritional recommendation also includes indications of activity level.

3 – Possible improvements throughout the text

In L148, put "classical" rather than "traditional".

Footnote 1 of page 4 seems to me to be very important from a methodological point of view: self-designation by the respondent rather than categories pre-defined by the interviewer. This should be better highlighted in the text itself.

Table 2 mentions N=2000 but the proportion of each ethnicity is unknown. It is necessary to see Appendix 2 for finding this information. It seems to me that it would be useful to add a column in Table 2 to give this indication.

In L226, modernization and urbanization are linked to no-breakfast and westernized forms. This seems to me to raise a question about the future of national identity: The only sources of modernity being exogenous, we imagine that those who still have breakfast come from lower social categories (less "modern"). Is this the case and is there not a point here to return to in final discussion and conclusion?

In Table 3, the term "typical" is reserved for foods corresponding to the respondent's ethnicity and "other typical" for food corresponding to ethnic groups other than that of the respondent. These influences between ethnic groups would probably deserve to be discussed at a minimum. Paradoxically, Nasi Lemak seems to be adopted more by the Indians than by the Malays themselves. And it is crossing all ethnies of the Malaysia society.

In L316, the authors express a "position", which is similar to a conviction more assertive than demonstrated: breakfast in Malaysia is a "nested identity practice". My feeling is that this assertion is core in the article. This is quite unusual in a scientific article to put such a core assertion in the middle of the results. I think it would be possible to make it a hypothesis higher in the text (no clear hypothesis is proposed) before presenting the results. And thus, to show the evidence that has been gathered for the benefit of this assertion. This is only a suggestion, but it seems to me to deserve the attention of the authors.

Author Response

The authors thank the reviewer for the time spent in reading and the numerous, abundant, and rich comments. They propose to answer the best while not being sure to be able to integrate all the suggestions and invitations for extension proposed by the reviewer.

Beyond the suggestions and remarks - which they will attempt to make the most of - they understand from in this review of more than 2000 words, the interest in the questions raised by a multidisciplinary perspective which go far beyond the framework of an article.

Comments are thus addressed in three ways. First, some suggestions call for simple and concrete answers, leading to modifications of the text. They relate to points that are insufficiently clear to readers and are detailed in the table below.  

Second, some comments are questioning the argumentative postures. For example, should the article be written with hypotheses formulated as such and data that attempt to verify them? In brief, is it necessary to write the article according to a deductive plan? The data from the MFB offers the possibilities of both hypothetic-deductive or inductive approaches. Here, the authors have chosen an inductive presentation as it is because they have followed this approach. They started investigating the subject with a few research questions, and the problematic was constructed at the same time as the data was analysed. In that case, the answer could be a justification. 

Finally, some remarks raise interesting questions which are going far beyond the scope of an article. On the one hand, the questions of anomy and the relationship between the sociology of food and public health are already the subject of a large literature by some of the authors of the article, largely in French, but also in English. The authors thank the reviewer for these openings and suggest expanding the discussion started with this article’s review, either within a seminar to which they will be pleased to invite him/her or - why not? – for a new special issue of the journal.

Questions, remarks, and suggestions

Answers from authors

1 - The Nasi Lemak, a central product in the demonstration.

Nasi Lemak seems to play a central role in this work because it is cited as early as L125. However, it is not really presented in its specificities and that is a pity. Possible variations in its preparation (ingredients, cooking methods) and their consequences on its nutrient composition should be explained. In addition, the text points out (L248) a controversy which, on reading, sheds light on the subject, but unfortunately, nothing more is said about this controversy. It is a major element of the whole article: the authors should introduce it correctly (and not stealthily in the middle of the results on the Malaysian style) and inform us about it. Who wears it, how it is expressed, is the fat content of Nasi Lemak essential for authenticity and taste, is its consumption correlated with the appearance of cases of obesity and type II diabetes, etc.? I recommend focusing on this emblematic food since it crystallizes the tensions studied.

The description of Nasi Lemak is added in note 3.

Another footnote (number 5) has been added. It reads as “The consumption of Nasi Lemak on a regular basis was pointed as a potential cause of overweight in Malaysia and sparked a debate. Some nutritionists were stressing on the quantity of fat and calories - a regular portion of nasi lemak with a boiled egg, sambal, and chicken accounts for approximately between 500 and 700 calories, 13 to 15 grams of protein, more than 14 grams of fat and more or less 80 grams of carbohydrates. Other voices echoed by TIME magazine in 2016, listed nasi lemak as one of ten most healthy international breakfasts. We can add that the question is difficult to arbitrate as the composition (and therefore the nutritional intake) varies considerably from one place of purchase to another.”

In addition, the question posed (in rhetorical form) in L250-251 on fat and the possibilities of reducing it to improve its nutritional profile, is formulated without treatment modality. Attributed to the Malay community ("our interest"), this question seems to be a major result of the article. However, it is not developed at all in the rest of the article and the authors do not return to this point in the discussion and in the conclusion. This is a pity because it would give real grips on how the debates (desired in the last sentence of the article) could be fueled. I recommend making it an axis of the final discussion by giving ways of dealing with how to overcome the contradictions identified.

The following paragraph has been added in lines 261-266 and reads as follows: “However, this question opens some issues in the Malaysian context. Firstly, nasi lemak can be sourced from diverse types of stakeholders in the food system. To revise the nutritional profile, it will be mandatory to work with the owners of restaurants and mamaks, as well as with all small traders involved in the more or less informal economy. How to access these actors of the food system? And secondly, how to ensure acceptance of the reprofiling of dishes with such emblematic meanings?”

2 – Dissonance and anxiety, link with obesity: fragile affirmations.

The text reveals in L282 the sources of anxiety related to the “dissonances” between the declarative and the actual practices. Figure 3 is supposed to provide the reader with something to identify the proportions of individuals concerned in each ethnic group. It remains difficult to understand what this is all about. I had to consult the cited reference (Fournier et al, 2016, reference 19) to better understand what was presented. Do the authors really need these clarifications in the article when the readers need other elements to better understand the point?

In addition, this reference 19 tells us that cases of elevated BMI are not correlated with dissonance (“obesity is over-represented in people who have dissonant eating behaviors”) and that, therefore, this direction is not necessarily the one to be explored. Social norms and practices are often at odds, but it does not seem obvious to make them a causality of obesity cases.

There remains “anxiety”, which, according to the authors, is an important factor to consider. This remains a possibility, however not demonstrated by the data collected, and which would require open interviews with the persons concerned. It seems to me that this should be expressed with caution in absence of reliable data.

A relevant link is made at the end of the article (L413) with the risks of anomia and the need to articulate nutritional recommendations and socio-cultural practices. However, it does not require, in my opinion, to go through the individualized aspects of anxiety that are not at the heart of the article.

The concept of dissonance calls too many theoretical references and explanations to be fully addressed in this paper only. Therefore, the authors have change cognitive dissonance to discrepancy (lines 289 and 291).

The dissonance in the ref 19 is not for breakfast.

Other references, for example in French Polynesia, clearly show the link between dissonance and obesity: Poulain J.P., « Combien de repas par jour ? Normes culturelles et normes médicales en Polynésie Française », Journal des Anthropologues, Norms to drink and to eat. Production, transformation and cunsumption of food norms, 2006, 245-268.

A first version of the paper included data that showed some statistical links. However, the authors have chosen to remove those as 1) it opens a debate that we have decide to keep for another article; 2) the small number of respondents in this scenario leads to statistical discussion. It is thus more of an indicator that invite to continue the exploration.

L 289 the formulation of the authors was already careful, i.e. “These gaps could potentially be a cause of dissonance and a source of anxiety”.

To reflect more cautiousness, we have decided to change the paragraph lines 293-298 to conditional. It now reads as follows: “This is the case, for example, of "counterproductive" effects that could be the consequence of food anxiety following the dissemination of messages which are difficult to apply because they contradict the social norms of the country and culture concerned.”

3 – Possible improvements throughout the text

In L148, put "classical" rather than "traditional".

Footnote 1 of page 4 seems to me to be very important from a methodological point of view: self-designation by the respondent rather than categories pre-defined by the interviewer. This should be better highlighted in the text itself.

Table 2 mentions N=2000 but the proportion of each ethnicity is unknown. It is necessary to see Appendix 2 for finding this information. It seems to me that it would be useful to add a column in Table 2 to give this indication.

In L226, modernization and urbanization are linked to no-breakfast and westernized forms. This seems to me to raise a question about the future of national identity: The only sources of modernity being exogenous, we imagine that those who still have breakfast come from lower social categories (less "modern"). Is this the case and is there not a point here to return to in final discussion and conclusion?

In Table 3, the term "typical" is reserved for foods corresponding to the respondent's ethnicity and "other typical" for food corresponding to ethnic groups other than that of the respondent. These influences between ethnic groups would probably deserve to be discussed at a minimum. Paradoxically, Nasi Lemak seems to be adopted more by the Indians than by the Malays themselves. And it is crossing all ethnies of the Malaysia society

Revised, “traditional” changed into “classical” (line 149).  

The note is now included in the text (lines 156-163) which reads as follows: “Data collection tools, whether 24-hour dietary recall or multi-day (3 or 7) diaries, overwhelmingly use meal names as input to the questions. In the self-administered questionnaires, the names of the meals are recorded in the notebook and constitute structuring elements of the response and the recall. When the researchers using these tools claim “self-designation” by the respondent, they mean that it is the respondent who registers content (food and beverages) in the "breakfast" category. It is therefore not the same method as that used in the “Food Barometers” in which it is food intakes (food contacts) which are the input into the reconstruction of food days.”

Thank you for the suggestion, the number of respondents and proportions for each ethnic groups are added to table 2.

Modernization index has been constructed to reflect the transformations faced by the Malaysian society in terms of urbanisation, changes in the social stratification with emergence of a middle class (proxy: income levels), social mobility (proxies: education attainment and income dynamics), and socio-demographic transition (proxy: size of household). Therefore, it is meant to measure changes beyond those of the social stratification. Details the construction of the modernization index has been added in footnote 4 (page 6).

We have added “We can observe the highest consumption rate for nasi lemak is among the Indian community and not the Malay one.” (lines 244-245).  

In L316, the authors express a "position", which is similar to a conviction more assertive than demonstrated: breakfast in Malaysia is a "nested identity practice". My feeling is that this assertion is core in the article. This is quite unusual in a scientific article to put such a core assertion in the middle of the results. I think it would be possible to make it a hypothesis higher in the text (no clear hypothesis is proposed) before presenting the results. And thus, to show the evidence that has been gathered for the benefit of this assertion. This is only a suggestion, but it seems to me to deserve the attention of the authors.

Please refer to the above elaboration on deductive versus inductive scientific approach.